# Continuity and Overlap of Roles in Victims and Aggressors of Bullying and Cyberbullying in Adolescence: A Systematic Review

**DOI:** 10.3390/ijerph17207452

**Published:** 2020-10-13

**Authors:** Estefanía Estévez, Elizabeth Cañas, Jesús F. Estévez, Amapola Povedano

**Affiliations:** 1Department of Health Psychology, Universidad Miguel Hernández de Elche, 03202 Elche, Spain; eestevez@umh.es (E.E.); jesus.estevez01@goumh.umh.es (J.F.E.); 2Department of Education and Social Psychology, Universidad Pablo de Olavide, 41013 Sevilla, Spain; apovedano@upo.es

**Keywords:** bullying, cyberbullying, dual role, overlap, systematic review

## Abstract

The objective of this study was to conduct a systematic review of research focused on analyzing the overlap and continuity of the roles in victims and aggressors of bullying and cyberbullying, as well as the exchange of roles in both harassment dynamics in adolescents. Searches in the main electronic databases for studies published in the last 20 years identified 19 studies that fulfilled inclusion criteria. The findings of the studies analyzed were not homogeneous, however, the main conclusion of all of them, to a greater or lesser extent, was that there is a component of continuity or superposition in the roles of both forms of bullying. Some studies also found an exchange of roles, especially in the case of victims and cybervictims who decide to reprimand their aggressors in an online context, becoming in cyberaggressors too. It is necessary to continue investigating the coexistence of bullying and cyberbullying and its exchange in certain contexts and people, as well as whether they are part of the same phenomenon with a certain continuity, or if cyberbullying is another expression of traditional bullying. Future intervention programs focusing on traditional school bullying could also evaluate their impact in situations of cyberbullying among peers.

## 1. Introduction

Victimization and bullying among peers affect a growing number of children and adolescents at the international level [1,2]. In the school context, traditional bullying implies coercing, forcing, threatening, abusing, dominating, or intimidating others, in a hostile and repeated way, as well as having an imbalance of power between the bully and the victim [3,4]. On the other hand, intimidation through electronic technologies has increased among adolescents in the last decade [5,6]. This fact is worrisome because cyberbullying, like traditional bullying, also implies intentional, unjustified attacks carried out repeatedly on victims who cannot easily defend themselves, but is instead done through the use of computers, mobile phones, and other electronic devices [7,8]. This similarity is not the only one between traditional and cybernetic bullying, since they both involve intentional, repetitive, and hostile behavior intended to cause harm [9,10] in a situation of imbalance of power [11].

Nevertheless, both forms of harassment also manifest several characteristics that make them different from each other. For example, the imbalance of power in bullying can reflect differences in physical strength or social status, whereas in cyberspace, it can also reflect differences in technological competence between victim and aggressor [12]. Another evident differential feature is the context in which both types of harassment occur, at school or in cyberspace. This implies that victims of traditional bullying can disconnect once the school day is over, whereas cybervictims do not have a safe place to hide or flee from abuse [9]. Thus, bullying ceases once the victim leaves school, whereas cyberbullying ends when the aggressor (not the context) decides it [13]. Just like the context, time is another barrier that cyberbullying overcomes. Unlike traditional bullying, where aggressions are only possible during the time aggressor and victim are together (at school or on the way home), in cyberbullying, the abuse can be active 24 h a day, 7 days a week [14], also making it possible to premeditate the message of harassment [15]. In relation to duration, the harassment in the digital world can be “eternal” [16] and easily reach a large audience [17], unlike bullying, which occurs at a specific time and whose audience is limited to the people physically present where the episode occurs. In addition, the audience of cyberbullying can reproduce an offensive comment, spread a demeaning video, and translate their own opinion in a forum or web page, etc., promoting the harassment to spread quickly and giving it an exponential scope [13]. Finally, cyberbullying is comfortable for those who perpetrate, due to this conduct keeping the anonymity of aggressors. In this way, anonymity causes the cyberbully to not feel fear repercussions or punishment [18] and encourages them to continue to carry out impulsive and abusive behaviors they would not perform in face-to-face interactions [19]. This contrasts with situations of face-to-face bullying, in which most of the time victims and aggressors know each other [20].

These similarities and differences have raised questions in the scientific literature about whether bullying and cyberbullying constitute the same kind of aggressive behavior, with cyberbullying being a modern and electronic form of school bullying, or whether these two forms of aggressive behavior are different problem behaviors. Several studies have attempted to answer these questions by analyzing the psychosocial problems associated with both forms. Thus, some studies suggest that cyberbullying is closely linked to school bullying, possibly constituting an extension of it, since both phenomena are just as devastating for those who suffer it [21,22,23]. Contrarily, other studies indicate that cyberbullying is not reflect traditional bullying due to the particular characteristics of the former being associated with more negative consequences for victims than those of bullying in a school context [8]. However, research on adjustment problems derived from double victimization (traditional and online) or even a dual role of aggressor and cyber-aggressor is still in its beginning stage, and most of the studies yielding these kinds of analyses failed to control for the co-occurrence of bullying and cyberbullying. The works that have tried to deepen in this sense suggest that those involved in a dual role show more psychological adjustment problems, as they experience the negative effects of both roles [24]. This fact, together with recent preliminary results arguing that direct peer harassment is one of the most powerful predictors of cyberbullying [9,25], alerts about the need to determine whether these two forms of harassment have distinctive identities and development processes [26] or whether they are part of the same process of intimidation.

It should be noted that the problem of cyberbullying covers a vast field of investigation. Some authors state that studying the subject is a necessary undertaking in a setting in which the Internet and other digital technologies are increasingly present in daily life, especially among young people, a population in which aggressions through new information and communication technologies can have serious consequences in terms of their emotional adjustment, as well as potentially causing traditional bullying to spread to the cybernetic context [27]. This hypothesis, sustaining that problems in the school context can be transferred to and continue in virtual spaces, it is very interesting if we bear in mind that a large percentage of cyberbullying behaviors also occur among schoolmates [28]. Equally, it is plausible that anonymity in the cybernetic environment is a way for traditional aggressors with peer grudges to transfer their role to the cybernetic environment and avoid social disapproval [29]. Some authors have even observed an exchange of roles in the school and cybernetic contexts, as anonymity can encourage traditional victims to take revenge on their aggressors online, now adopting the role of cyberaggressors [26,30]. As can be appreciated, identifying the common factors of these two harassment behaviors is one of the great challenges facing scholars on the subject.

Although several studies have already explored the overlap between traditional bullying and cyberbullying, the results are often divergent, which makes it difficult to conclude whether cyberbullying is a distinct form of behavior or a variation or a new strategy of traditional bullying. Moreover, as mentioned earlier, many studies have investigated the problems associated with traditional bullying and cyberbullying; however, those do not measure the effect of being involved in both forms of bullying. The need to gather information in this regard is fundamentally based on the fact that it will lead to a better understanding of the dynamics of mistreatment among peers; the consequences for victims, aggressors, and victim-aggressors; and to defining more effective strategies of prevention and intervention in adolescents since they are quite vulnerable to these behaviors. Thus, the objective of this study was to conduct a systematic review of studies focused on analyzing the overlap and continuity of the roles in victims and aggressors of bullying and cyberbullying, as well as the exchange of roles in both harassment dynamics in adolescents.

## 2. Methods

The review was prepared following the PRISMA guidelines, for which definitions have been adopted from the Cochrane Collaboration. The purpose of these guidelines is to ensure that the articles included are reviewed in their entirety in a clear and transparent manner. Figure 1 shows the flow diagram with the four phases, recommended by the PRISMA guidelines, in which the inclusion/exclusion of each article is detailed.

### 2.1. Search Strategy

A systematic search of materials published in the last 20 years (from 2000 to present) was performed through consulting the following electronic databases: PsychInfo, Scopus, PubMed, and Web of Science. The search strategy was developed for each database using the combination of the terms “bullying” AND “cyberbullying” AND “victim*” AND “aggressor*” OR “bully” AND “cybervictim*” AND “cyberaggressor*” OR “cyberbully” AND “overlap” OR “dual role” OR “co-occurrence”. Initially, duplicates were removed from the total number of identified records. Abstracts from the remaining references were screened to retrieve full-text manuscripts. Finally, studies fulfilling inclusion criteria were selected for the assessment.

### 2.2. Inclusion and Exclusion Criteria

The search was limited according to the following inclusion criteria:
(1)Studies about bullying and cyberbullying, as well as victimization and cybervictimization in the school and the cybernetic context.(2)Studies whose aims (at least one) were to analyze the continuity and superposition of roles between bullying in the school and the cybernetic context and/or examine the exchange of roles from victim to aggressor and vice versa, as long as it was considered a form of superposition between both behaviors.(3)Studies in which the participants were adolescents enrolled in middle and high school or secondary education study centers.(4)Quantitative studies or scientific articles in which design was cross-sectional or longitudinal.(5)Papers in Spanish or English, due to difficulties in translating papers in other languages.

The exclusion criteria contemplated in the search were

(1)Studies investigating bullying in other contexts, such as reformatories, bullying among foster siblings, and bullying among children living in kinship care.(2)Studies investigating the continuity and superposition in other forms of abuse (e.g., domestic violence, urban violence, elder abuse, sexual abuse).(3)Studies involving infant or adult participants, as well as students in primary or university education.(4)Reviews, editorials, theoretical articles, grey literature, dissertations, books, case studies, and conference proceedings without conference papers available in the databases.(5)Papers in languages other than Spanish and English.

### 2.3. Study Selection Process

After compiling the manuscripts, we classified the studies, identifying those that met the inclusion criteria. For each of the studies, we extracted the following information: author and year of publication, study methodology, sample information, instruments for collecting data, key findings, and conclusions. These data were extracted by a researcher and verified by a second researcher to ensure the quality and accuracy of the information. Doubts or disagreements between evaluators were resolved through discussion and consensus with the help of a third reviewer. The results of this selection process are reported below.

### 2.4. Methodological Characteristics of the Included Studies

A modified version of the Quantitative Research Assessment Tool, developed by Child Care and Early Education Research Connections [31], was used to assess the methodological strength of the studies included in this review. This tool, which includes 12 items, was designed to provide general guidelines evaluating the quality of research studies. For this review, we selected five items of the tool (items 2, 3, 6, 9, and 10): “Randomized Selection of Participants”, “Sample Size”, “Operationalization of Concepts”, “Appropriateness of Statistical Techniques”, and “Omitted Variable Bias” (see Table 1). Item 1 (“Population”) was discarded because it was not considered relevant for the review, as all studies focused on a specific subset of the population (adolescents) and, as most of the studies were cross-sectional, we also excluded Item 4 (“Attrition Rate and Follow-up Studies”). Given that we considered the operationalization of the variables of each of the studies in order to analyze the methodological quality, we also discarded Item 5 (“Main Variables or Concepts”) in order to avoid redundant information. Finally, Item 7 (“Numeric Tables”), Item 8 (“Missing Data”), Item 11 (“Analysis of Main Effect Variables”), and Item 12 (“Research Ethical”) were deleted because they did not provide important information for the object of the present review.

In addition, two items of our own elaboration were added: “Frequency of bullying and cyberbullying” and “Descriptive analysis and reliability of the instrument”, whose purpose was to evaluate whether the selected studies considered the frequency with which the two harassment dynamics occurred and whether it carried out a descriptive and reliability analysis of each of the instruments administered. Finally, we set 7 criteria in the final version, which allowed us to verify the homogeneity of the studies, especially that of the instruments used, which was essential for the comparison of the results. Each item could be rated as −1, 0, 1, or NA, and thus the total score could range from −7 to 7. According to the specifications of the tool, studies with lower scores should be regarded with more caution compared with studies with higher scores. Eight of the studies reviewed had a score of 5 or higher.

As shown in Table 1, some studies used interviews or surveys and did not present reliability indices of the data collection instruments or did not establish categories according to the frequency of the harassment. These methodological aspects could explain, in part, the disparity in the percentages of continuity and overlap of the documented roles.

## 3. Results

Using the research strategy described above, we identified a total of 1838 references. After eliminating duplicates, 643 references were retained. Of these references, 33 were selected by title and abstract for reading full-text. Finally, 19 studies were included for meeting the inclusion criteria. Out of a total of 19 publications addressing this issue, 9 showed, to a greater or lesser extent, the existence of superposition and continuity between the roles of traditional bullying and cyberbullying [28,32,39,41,42,43,45,49], 5 found a role exchange between victims and offenders [34,35,37,38,50], and 5 observed both the continuity or overlap of roles in both phenomena and the exchange of roles [26,40,46,47,48]. It should be noted that the groupings were made on the basis of the conclusions of each of the studies, however, a percentage of victims and offenders adopted this role simultaneously both in the school and cybernetic context, or, in contrast, changed their role. The results of the different investigations can be seen in detail in Table 2.

### 3.1. Overlap and Continuity of Roles

Several studies suggest that, in general, being involved in bullying in the role of aggressor increases the risk of engaging in this role in the cybernetic context. For example, the results shown by Kim et al. [42], in a longitudinal study, revealed that similar patterns in the development process of both types of harassment (more than 60% of adolescents showed similar patterns), which indicated that both “traditional” and cybernetic bullies share key characteristics such as risk factors and protective factors, showing the similarity between the two problems. Del Rey et al. [36] analyzed the directionality of the roles and concluded that those involved in traditional harassment are more likely to extend their behavior to the cybernetic environment than conversely—that is, those involved in cyberbullying are less likely to begin to develop traditional bullying behavior at school. In this sense, Baldry et al. [32] showed in their study that traditional bullies were almost five times more likely to be cyberbullies than those who did not bully at school.

The overlap and continuity between traditional bullying and cyberbullying are also observed in the role of victims, with findings showing that victims of traditional bullying may be more likely to experience cyberbullying attacks than non-victims [26]. Baldry et al. [32] analyzed the probability of being a victim in traditional and cybernetic bullying simultaneously, finding that victims of traditional harassment were almost four times more likely to be cybervictims than non-victimized students. These results are in line with those found by Juvonen and Gross [39] and Khong et al. [41] who found that being a victim at school increased the risk of being bullied online in almost 7 and 11 times, respectively. Following these results, some investigations studied how many cybervictims had also suffered victimization at school. Beran and Li [33] found that 30% of the cybervictims from their study also experienced traditional victimization. Along the same line, in the works of Wang et al. [48] and Khong et al. [41], the rates of cybervictims who reported to have experienced victimization at school increased to 48.7% and 68.9%, respectively.

Some authors observed that approximately half of the students identified as victims were found simultaneously in both contexts [28,43,45]. Although only one-ninth of the population was identified as victims in both contexts in the study of Schneider et al. [36], the result is still concerning. These results are in line with those found by Katzer et al. [40], who confirmed that aggressors usually attack their victims in both contexts, both directly and through new technologies. Regardless of being a bully or victim, Kubiszewski et al. [43] observed that the fourth part of the students kept the same role in both forms of harassment.

### 3.2. Exchange of Roles as a Form of Continuity

Some of the studies reviewed concluded that there is sometimes an exchange of roles between victims and aggressors of traditional bullying and cyberbullying. In particular, it has been observed that some victims of traditional bullying can change to the role of aggressor in cyberspace because the online context provides them a means that they perceive as safer and where they can take revenge on their aggressors [26,33,35], hide their identity [50], and change the balance of power [40]. Indeed, in the works of Slonje and Smith [46] and Caudrado-Gordillo et al. [35], the authors concluded that approximately 10% of traditional victims commit aggression in cyberspace.

Concerning this possible exchange of roles, or “overlap of aggression and victimization” in the same context, Gradinger et al. [38] identified that students who were victims and bullies in terms of traditional harassment were overrepresented (32.7%). In addition, Cuadrado-Gordillo et al. [34] found victims of bullying in traditional or virtual environments tended to use the same means to bully their peers; however, they also detected a strong correlation between being subjected to traditional bullying situations and choosing to attack their peers through online means. Similarly, Katzer et al. [40] observed that chat victims demonstrated their own bullying behavior exclusively in the environment of their victimization, which could be interpreted as a way of “fighting back” or “letting off steam”. In this sense, other studies suggest that the exchange of roles tended to happen in the opposite context. This was the case of the work of Wang et al. [48], in which it was observed that bully-victims who act in combined bullying were more numerous (17.1%) than those who were only involved only in traditional (12.1%) or cybernetic (5.4%) contexts.

In this game of exchange and continuity of roles, Del Rey et al. [36] concluded that it is more likely a victim becomes a bully than a bully becomes a victim. In line with this, Beran and Li [33] provided another very interesting idea involving the possibility that traditional victims, who use cyberspace as a means of attacking their aggressors, may provoke an act of response from the aggressor (now also cyberaggressor), extending, in consequence, their victim status to the online context.

## 4. Discussion

The present study offered a systematic review about the overlap and continuity of the roles in victims and aggressors of bullying and cyberbullying, as well as the exchange of roles in both harassment dynamics. On the basis of the literature included in this review, the results suggest that, in general, there is a component of continuity or overlap in the roles of both harassment dynamics.

Several studies of the present review have shown that a high percentage of young people involved in some type of bullying (traditional or cybernetic) experience or increase the probability of experiencing the same role in the other context [27,32,33,36,39,40,41,42,43,45,47,48,49,50]. According to previous literature, some researchers have explained this fact as being due to the similarity and coincidence in the risk and protection factors detected both for bullying and cyberbullying as reasons for the existence of overlapping roles, these being the similarity and coincidence in the risk and protection factors detected both for bullying and cyberbullying [9,32]. Another argument is based on the observation that being a victim or aggressor of traditional bullying is precisely a risk factor for being abused or being a bully in the cybernetic environment [28,40,41,42,46]. That is to say, cyberbullying rarely occurs in the absence of traditional bullying [39]. In this sense, the cybernetic environment could be interpreted as a forum that extends the school grounds [42], where the home is no longer perceived as a safe place, and victimization and intimidation are available 24 h a day [8]. If this were the case, following the argument of some authors, the interventive effect of prevention programs in traditional bullying might, in turn, have an impact on the decrease of cyberbullying [30].

Although the results of the research analyzed showed, to a greater or lesser extent, a common conclusion, the rates of overlap of each of them were not homogeneous. Variability across studies may be because some studies considered diverse displays of bullying and cyberbullying while others used a general measure of both dynamics. Moreover, measuring the frequency of both forms of harassment through different periods of time could also be the reason of the variability between studies. This may also explain the notable percentage of adolescents is involved in a single role and type of harassment [45,49]. Thus, there is a relevant percentage of cyberbullies who only behave violently with their peers through the network, possibly because of the anonymity that this type of harassment allows them, and the disinhibition that the Internet and new technologies encourage in young people [43]. These peculiarities lead youth to engage in bullying behaviors that they would otherwise avoid [49].

Despite the heterogeneity mentioned, findings of the present study showed that sometimes adolescents manifest different roles that depend on the context of harassment in which they are involved [27,32,33,34,35,37,38,40,46,48,50]. The exchange of roles can also be understood in some sense as an overlap between bullying and cyberbullying. The decision of traditional victims or cybervictims to assault their aggressors in cyberspace turns them into cyberaggressors [26,35], thus promoting the coexistence of school and online abuse in the same person [45]. This exchange of roles has been interpreted mainly as a way to counteract the consequences associated with the victim role [34,35]. In other studies, the exchange of roles has been understood as a complex process that varies depending on the perception of the harassment or the end sought through it. For example, students intimidated at school who conclude that aggression towards peers provides certain social benefits (such as acceptance in a group) may decide to abuse others in cyberspace in order to be socially accepted [48]. Another possibility is that students may view cyberspace as a way to confront their aggressors without exposing themselves directly to them [26,30], thus avoiding likely reprisals [49]. However, it is very likely that the victim receives a cyberattack by the bully, and consequently extends the initial traditional victimization to the cybernetic environment [48].

### Limitations and Considerations for Future Research

The exposed findings are not exempt from limitations. The first limitation arises from the excluding criteria since it might constitute a bias for the results. Following a stricter protocol and criteria would have made the contribution of this review more robust. Furthermore, it only consulted a limited number of psychology databases. Perhaps consulting a larger number of scientific databases relevant to further disciplines might have strengthened the contribution of this work. Another limitation is due to the experimental designs of studies evaluated. In most of them, the instruments used do not match, and in other studies bullying and cyberbullying were measured through questionnaires that were not validated. In addition, most of the studies were not considered methodologically sound due to the low qualification obtained in several methodological criteria. This contributed to the impossibility of performing meta-analyses due to the scarcity and heterogeneity in quantitative data.

Despite modest conclusions, this work has provided an understanding of the overlap between traditional and cybernetic harassment. However, the authors are aware that much more work needs to be done. Thus, for example, for future studies, it would be convenient that further reviews include studies with good methodological quality, rigorous eligibility, and sample selection criteria, using instruments and measures that have been previously validated in the literature for investigation of the overlap between different contexts of bullying to strengthen the evidence about this important issue.

## 5. Conclusions

Although there are limitations in the present review, we nonetheless offer an extensive picture of the phenomenon of peer violence in traditional and cybernetic contexts. The analyzed studies in this review show that, although there is a certain continuity and overlap between both forms of harassment, it is difficult to reach a consensus. However, it is known that being involved in multiple forms of bullying and victimization at the same time increases the risk of adjustment problems to every one of the involved parties [37]. For example, cyberspace could be interpreted, for victims who were bullied in school and cybernetic contexts simultaneously, as an extension of the school environment, available 24 h a day; for traditional victims, as a place to assert dominance over others as compensation for being harassed at school; and for cyberbullies, as a place where they take a more aggressive personality. Thus, it is necessary to continue investigating the coexistence of bullying and cyberbullying and their exchange of roles and contexts, as well as whether they are a continuity of the same phenomenon, or whether cyberbullying is another expression—a subtype—of traditional bullying. The set of common and specific characteristics of each type of harassment shows the importance of designing future works that jointly analyze the dual roles, as well as prevention and intervention programs that consider both types of bullying or evaluate whether the programs of traditional bullying may also have an impact in situations of cyberbullying among peers.

## Figures and Tables

**Figure 1 ijerph-17-07452-f001:**
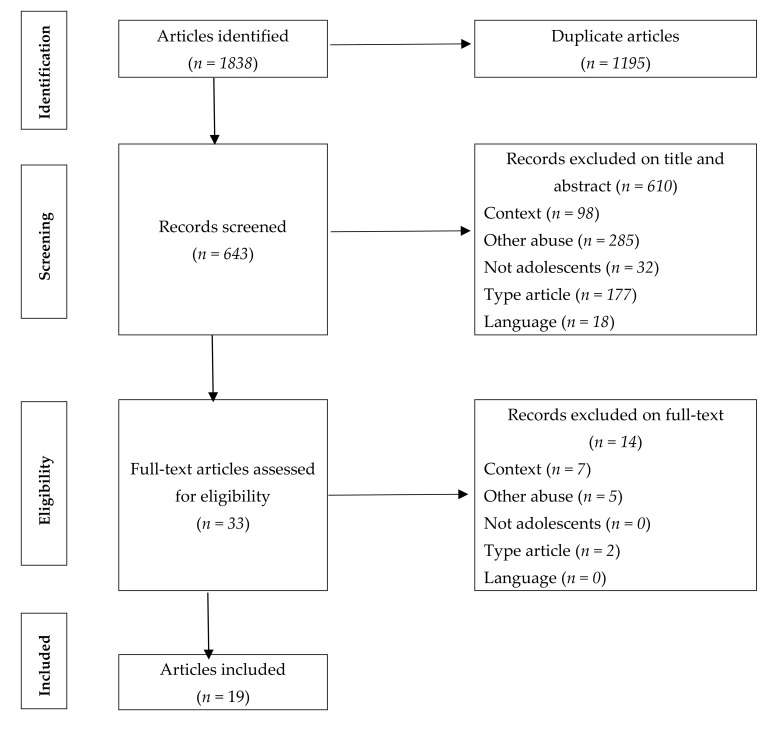
Flow chart of the selection process. **Exclusion criteria: Context:** studies investigating bullying in other contexts; **Other abuse:** studies investigating other forms of abuse; **Not adolescents:** studies not involving adolescent participants; **Type article:** non-quantitative studies or scientific articles; **Language:** study not written in English or Spanish.

**Table 1 ijerph-17-07452-t001:** Methodological quality of studies.

Author/s	Criteria							Total (−7 to 7)
	I	II	III	IV *	V *	VI	VII	
Baldry, Farrington, and Sorrentino [32]	0	0	1	1	1	1	1	5
Beran and Li [33]	0	−1	0	0	−1	1	1	0
Cuadrado-Gordillo and Fernández-Antelo [34]	1	0	1	1	1	1	1	6
Cuadrado-Gordillo, Fernández-Antelo, and Martín-Mora [35]	1	0	1	1	1	1	1	6
Del Rey, Elipe, and Ortega [36]	0	−1	1	−1	1	1	1	2
García-Fernández, Romera-Félix, and Ortega-Ruiz [37]	1	0	1	−1	0	1	1	3
Gradinger, Strohmeier, and Spiel [38]	1	1	1	−1	1	1	1	5
Juvonen and Gross [39]	0	0	0	1	0	1	1	3
Katzer, Fetchenhauer, and Belschak [40]	1	0	1	1	1	1	1	6
Khong et al. [41]	1	0	1	1	0	1	1	5
Kim, Song, and Jennings [42]	−1	0	0	0	−1	1	1	0
Kubiszewski, Fontaine, Potard, and Auzoult [43]	1	0	1	−1	−1	1	1	2
Lazuras, Barkoukis, and Tsorbatzoudis [44]	1	0	1	−1	−1	1	1	2
Schneider, O’donnell, Stueve, and Coulter [45]	1	1	1	−1	−1	1	1	3
Slonje and Smith [46]	1	−1	0	−1	−1	1	1	0
Waasdorp and Bradshaw [47]	1	1	0	−1	−1	1	1	2
Wang et al. [48]	1	0	1	1	1	1	1	6
Ybarra, Diener-West, and Leaf [49]	1	0	0	1	1	1	1	5
Ybarra and Mitchell [50]	1	0	0	−1	1	1	1	3

Notes: * Own elaboration; I. Randomized Selection of Participants: (1) random selection, (0) nonrandom selection, (−1) no description of the sample selection procedure, (NA) not applicable; II. Sample Size: (1) sample size larger than similar studies, (0) sample size the same as similar studies, (−1) sample size smaller than similar study or sample size not given, (NA) not applicable; III. Operationalization of Concepts: (1) variables have either been previously used in research or are improvements over previous measures, (0) variables have not been used in previous research studies, (−1) variable operationalization is not discussed, (NA) not applicable; * IV. Frequency of Bullying and Cyberbullying: (1) categories are established according to frequency with detailed and validated procedure, (0) categories are assigned according to frequency without specifying the procedure to reach them, (−1) the frequency of harassment and cyberbullying is not taken into account, (NA) not applicable; * V. Descriptive Analysis and Reliability of Instruments: (1) the instruments are described and their reliability index is set, (0) some type of description of the instrument is presented but not the reliability of this one, (−1) the instruments are not described and do not present reliability, (NA) not applicable; VI. Appropriateness of Statistical Techniques: (1) statistical techniques, reasons for choosing technique, and caveats are fully explained, (0) statistical technique is explained, but the reasons for choosing technique or the caveats are not included, (−1) statistical technique, reasons for choosing technique, and caveats are not explained, (NA) not applicable; VII. Omitted Variable Bias: (1) all important explanations are included in the analysis, (0) important explanations are omitted from the analysis, (−1) variables and concepts included in the analysis are not described in sufficient detail to determine whether key alternative explanations have been omitted, (NA) not applicable.

**Table 2 ijerph-17-07452-t002:** Summary of selected studies.

Author/s	Characteristics of the Sample	Self-Report Measures and Objectives	Main Findings
Baldry et al. (2016) [32]	*N* = 5058 (53% girls; 47 % boys) Age range = 11–18 Italy	Different types of bullying and cyberbullying to analyze the overlap between both behaviors.	Bullies were almost five times more likely to become cyberbullies and victims were almost four times more likely to be cybervictims. Traditional victims tend to use online space to start (cyber)bullying others and invert their role (role exchange).
Beran and Li (2008) [33]	*N* = 432 (55% girls; 45% boys) Age range = 12–15 Canada	Experience of cyberbullying and bullying to determine whether traditional victims are also cybervictims.	A third of the children of the sample who were bullied in cyberspace were also bullied at school (continuity). There is also a role exchange in cybervictims motivated by personal motivations (role exchange).
Cuadrado-Gordillo and Fernández-Antelo (2014) [34]	*N* = 1648 (48.9% girls; 51.1 % boys) Age range = 12–16 Spain	Different types of bullying and cyberbullying to identify the role of victim-cyberaggressors and their prevalence.	Victims of bullying in school or virtual environments tended to use the same means to bully their peers, but also they chose to attack their peers through online means (role exchange).
Cuadrado-Gordillo et al. (2019) [35]	*N* = 1648 (48.9% girls; 51.1 % boys) Age range = 12–16 Spain	Different types of bullying and cyberbullying to determine the prevalence of victim-aggressors in both contexts.	Suffering victimization can predict the aggressive response of many of the adolescents due to the anonymity that can be achieved by using technological and virtual resources (role exchange).
Del Rey et al. (2012) [36]	*N* = 274 (48% girls; 52% boys) Age range = 12–18 Spain	Experience of cyberbullying and bullying in two different time periods to analyze the homogeneity or exchange between the roles of both harassment dynamics.	Bullying participation helped to predict cyberbullying participation. Traditional victim role could predict victimization and cybervictimhood in the future (continuity).
García-Fernández et al. (2015) [37]	*N* = 1278 (47% girls; 53% boys) Age range = 10–14 Spain	Experience of cyberbullying and bullying during the last 3 months to study the overlap between both behaviors.	Being involved in cyberbullying problems seems to be a factor related to involvement in traditional bullying problems. A consistent part of adolescents was involved in bully and victim roles at the same time (role exchange).
Gradinger et al. (2009) [38]	*N* = 761 (52% girls; 48% boys) Age range = 14–19 Austria	Different types of (cyber)bullying and (cyber)victimization to investigate the co-occurrence of these behaviors.	Hardly any students were exclusively cybervictims, with most of them being traditional victims at the same time. Students in traditional bully-victim or combined bully-victim groups were overrepresented (role exchange).
Juvonen and Gross (2008) [39]	*N* = 1454 (75% girls; 25% boys) Age range = 12–17 USA	Experience of cyberbullying and bullying and assumptions about cyberbullying to examine the overlap and the similarities between online and traditional bullying among Internet-using adolescents.	Being bullied in school could be a risk factor for being bullied online (continuity). Anonymity did not support the assumption school-based victims using cyberspace to retaliate, due to cyberbullied youth being more likely to retaliate in school than online (role exchange).
Katzer et al. (2009) [40]	*N* = 1700 (55% girls;45% boys) Age range = 11–17 Germany	Experience of cyberbullying and bullying to determine the differences or similarities in the predictors of both harassment dynamics.	Traditional victims also tended to be cybervictimized (continuity). Victims tended to perpetrate bullying towards others in the school context and cybervictims in cyberspace, becoming a victimized aggressor (role exchange).
Khong et al. (2020) [41]	*N* = 3329 (49.8% girls; 50.2 % boys) Age range = 12–17 Singapore	Experience of cybervictimization and victimization to examine the co-occurrence of both dynamics.	Victims of bullying were almost 11 times more likely to be cybervictims, compared to those who had not experienced school bullying (continuity). Cyberbullying rarely occurs in the absence of traditional bullying, being part of a larger bullying pattern.
Kim et al. (2017) [42]	*N* = 2721 (50% girls; 50% boys) Age = 14 South Korea	Experience of cyberbullying and bullying during the last year to study the differences and similarities between both harassments.	Bullying increased the risk of being a cyberbully and vice versa. Predictive and protective factors of bullying and cyberbullying were similar.
Kubiszewski et al. (2015) [43]	*N* = 1422 (43% girls; 57% boys) Age range = 10–18 France	Experience of cyberbullying/victimization and bullying/victimization during the last 2–3 months to study the overlap of these dynamics.	The fourth part of the students kept the same role in both harassments (little overlap).
Lazuras et al. (2017) [44]	*N* = 1004 (51% girls; 49% boys) Mean age = 14.88 Greece	Experience of cyberbullying and bullying during the last 2 months to examine the overlap between both forms of harassment.	Bullies tended to harass in cyberspace also and traditional victims were more likely to become cybervictims (continuity). Some school victims participated as cyberbullies (role exchange).
Schneider et al. (2012) [45]	*N* = 20.406 (51% girls; 49% boys) Age range = 14–17 USA	Experience of cyberbullying/victimization and bullying/victimization in the last 12 months to examine the degree of overlap between these behaviors.	Almost two-thirds of all cybervictims reported that they were also harassed at school and, conversely, more than a third of victims reported that they were also harassed in cyberspace (continuity).
Slonje and Smith (2008) [46]	*N* = 360 (44% girls; 56% boys) Age range = 12–20 Sweden	Experience of cyberbullying and bullying during the last 2–3 months to determine whether cyberbullying is a subtype of bullying.	A small percentage of traditional victims reported cyberbullying others (role exchange). Cyberbullying was considered independent to asserting dominance over others as compensation for being bullied at school.
Waasdorp and Bradshaw (2015) [47]	*N* = 28.104 (49% girls; 51% boys) Age range = 14–17 USA	Different types of bullying and experience of cyberbullying in the past 3 months to examine the overlap between bullying and cyberbullying.	More than half of the victims were also cybervictims. Cyberbullying is a risk in terms of experiencing other forms of bullying in turn (continuity).
Wang et al. (2019) [48]	*N* = 2111 (51.4% girls; 48.3% boys) Age range = 14–20 Taiwan	Experience of cyberbullying/victimization and bullying/victimization to investigate the correlates among these profiles.	A third of adolescents were traditional bully-victims and almost a third of adolescent were cyberbully-victims. A significant number of students reported being victims of traditional and cyberbullying simultaneously (continuity).
Ybarra et al. (2007) [49]	*N* = 1588 (47% girls; 53% boys) Age range = 10–15 USA	Experience of cyberbullying/victimization to know whether cyberbullying is an extension of bullying.	A third of the cybervictims were also victims at school (continuity). The rate of internet harassment was similar for youth who are home-schooled and youth who are schooled in public/private schools, suggesting that it is not always an extension of school bullying.
Ybarra and Mitchell (2004) [50]	*N* = 1.501 (69% girls; 31% boys) Age range = 10–17 USA	Experience of cybervictimization and victimization simultaneously in the last year to expand knowledge about both roles.	Half of the cyber-victims and cyber-aggressors were also victims and traditional aggressors, respectively. Almost a third of youth were found to be exclusively involved in harassment online (continuity).

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
