# Peer review of "Continuity and Overlap of Roles in Victims and Aggressors of Bullying and Cyberbullying in Adolescence: A Systematic Review"

_ijerph, 2020, doi:10.3390/ijerph17207452_

Round 1

Reviewer 1 Report

Dear Authors,

Your work does a compilation of the last research in this field intending to analyze the overlap and continuity of the role in victims and aggressors of bullying and cyberbullying, as well as, the exchange of roles in both harassment dynamics. The article is fine, although you have to improve some things on methods and results sections. I write you some questions:
In the methodology section, you do not explain the method you apply (Ej. Cochrane, other in education), neither you demonstrate how you built your research question (ej. PICO, ? ).
Likewise, the criteria for inclusion and exclusion is confusing, you must go deeper in detail. It would be better if you structure and define better this section.
The Study Selection process and the results are mixed. The flow chart is in the results section, but it is the process. You can improve this.
Likewise, You have to include in the introduction section more updated references.
I trust you can improve your work.

Author Response

Dear reviewer,

We appreciate your comments. In relation to your suggestions we have made the following changes:

- We have included in the methodology section the method applied for the inclusion and exclusion of the results.

- We have written the justification of the study more clearly, which has given rise to the research question of this work.

-We have rewritten the inclusion and exclusion criteria in a clearer way.

- We have moved the flow chart to the section of “Study selection process”.

- We have included Introduction new up-to-date references such as:

Martínez, J.; Rodríguez-Hidalgo, A. J.; Zych, I. Bullying and cyberbullying in adolescents from disadvantaged areas: validation of questionnaires; prevalence rates; and relationship to self-esteem, empathy and social skills. Int. J. Environ. Res. Public Health 2020, 17, 6199.

Savage, M. W.; Tokunaga, R. S. Moving toward a theory: Testing an integrated model of cyberbullying perpetration, aggression, social skills, and Internet self-efficacy. Comput. Hum. Behav. 2017, 71, 353-361.

Sung, Y. H. Book review of cyber bullying approaches, consequences and interventions. International Journal of Cyber Criminology 2018, 12, 353-361.

Subrahmanyam, K. (2019). Online Behaviors. In The Encyclopedia of Child and Adolescent Development, Jewell, J. D., Hupp, S., Eds.; John Wiley & Sons, Ltd: USA, 2019; pp. 1-10.

Reviewer 2 Report

Thank you for the opportunity to review the above manuscript. I really enjoyed reading it. The authors did a very good job pulling together the various studies and evaluating them in terms of methodological strength before inclusion. My inly criticism is in terms of the implications of the findings of the systematic review. I think these could be strengthened in the paper.

Author Response

Dear reviewer,

Thank you very much for your comments.

In relation to the population of the study, we have specified in the title and abstract the population for which this study is intended. However, the authors decided that the population (adolescents) should not be entered into the search strategy to avoid that exploration deleted relevant studies that used other terminology to refer to adolescents (youth, teenagers…).

Reviewer 3 Report

The current study provided a systematic review of research on the overlap and continuity of the roles in bullies and victims, in the realms of bullying and cyberbullying. I have several concerns about this study, which are detailed below. 

The study only reviewed empirical research on adolescents only (according to inclusion criteria). It’s confused that the study title, abstract, and even the search strategy did not explicitly mention this point. The authors should address the study focused on this specific population at the beginning.

Introduction

Main problem: Introduction is vague and not easy to follow. Authors are suggested to merge several short paragraphs into one, with a guiding sentence at the beginning.

The study proposed a hypothesis, but how did it derive from? What’s the theoretical background?

As for the research gaps, hard to say “very few”. Bully-victims have long been discussed and lots of research has explored the dual role of victims and bullies. The authors are suggested to revise the sentence or narrow down the specific gap.

Method/Results

Main problem: the authors used qualified measurement to assess the quality of reviewed studies, which is a strength. However, there is no further exploration on the methodology features or limitations in the results. Implications for future research is suggested to add in the discussion.

Main problem: Table 2 is too long and too crowded. No need to include information like title (already in the reference list). Too many words in objective/results/findings and readers won’t have time to go through them. Instead, merge author/year into one column, merge objective/results/findings together and only present key information using phrases, keywords, short sentences, etc. It not a combination. The authors are suggested to shorten the column as much as possible.

Method details should be presented clearly, such as any theory framework, any difference/similarities in the measurement of bullying/ cyberbullying, the recall period (over the past year, past months, or ever), self-reported or teacher/parents reporting, etc.

Please refer to tables in the following review studies:

Espelage, D. L., Valido, A., Hatchel, T., Ingram, K. M., Huang, Y., & Torgal, C. (2019). A literature review of protective factors associated with homophobic bullying and its consequences among children & adolescents. Aggression and violent behavior45, 98-110.

Mazzone, A., Nocentini, A., & Menesini, E. (2018). Bullying and peer violence among children and adolescents in residential care settings: A review of the literature. Aggression and violent behavior38, 101-112.

Discussion

Main problem: Discussion is descriptive. Should summarize the main findings, by topics, methods, etc. and add limitation and implication for future research.

It should be noted that the prevalence of bullying and victimization are hard to compare across studies, because of different age range, measurement, cultural context, etc.

Author Response

Dear reviewer,

We appreciate your comments.

In relation to your suggestions, we have made the following changes to improve the work:

Regarding the introduction:

- We have merged and redrafted several paragraphs of the introduction so that it can be read more clearly.

- We have deepened the justification of this study to clarify the hypothesis and theoretical background of the study.

- We have revised and modified the sentence that referred to “research gaps”.

Regarding the method / results:

- We have included a paragraph of limitations (in the discussion section) in which we highlight methodology features or limitations in the results.

- We have reduced Table 2, keeping the most relevant information for readers.

- We have improved the writing of the method, by presenting the information in more detail.

Regarding the discussion:

- We have summarized the main findings and they have been contrasted with previous research.

- We have highlighted the limitations of the study and incorporated a section with considerations for future research.

Round 2

Reviewer 1 Report

Dear authors,

The article is better than before. However, in the article there are some change that you have to modify or clarify, these are:

  • Criteria inclusion/exclusion: It is confusing.  You have to review the criteria because of there are gaps. For example, in the exclusion criteria, you say this: "(1) Studies investigating bullying in other contexts, such as reformatory, and bullying among foster siblings and among children living in kinship care." Nevertheless, in the inclusion criteria, you don't explain the context you included. 
  • Flow chart: This figure is not right. There are some errors such as arrows. Also, if it is possible, you should change the word of "Exclusion 1, 2..." to a short name of exclusion. For example, Context, etc.  

Likewise, I advise you to review the document, such as to justify and capital letters. It's little. 

Author Response

Dear reviewer,

We appreciate your comments.

In relation to your suggestions, we have redefined the inclusion / exclusion criteria so that both are now more coherent.

We have also corrected the arrows in the flow chart and have substituted the word "Exclusion" for a term consistent with the exclusion criteria it refers to.

Finally, we have justified the paragraphs that were aligned to the left and revised the capital letters.

Thank you very much.

Reviewer 3 Report

The authors have made constructive and thorough efforts to address the concerns raised by the reviewer. The paper has been improved substantially. 

Please check table 2, column 3 of Cuadradon-Gordillo & Fernández Antelo (2014). The first and second sentences seem duplicate.

Author Response

Dear reviewer,

We really appreciate your comments.

In relation to your suggestion, we have removed the duplicate phrase from Table 2.

Thank you very much.